# Arterial Aneurysm Localization Is Sex-Dependent

**DOI:** 10.3390/jcm11092450

**Published:** 2022-04-27

**Authors:** Daniel Körfer, Caspar Grond-Ginsbach, Maani Hakimi, Dittmar Böckler, Philipp Erhart

**Affiliations:** 1Department of Vascular and Endovascular Surgery, Heidelberg University Hospital, 69120 Heidelberg, Germany; caspar.grond-ginsbach@med.uni-heidelberg.de (C.G.-G.); dittmar.boeckler@med.uni-heidelberg.de (D.B.); philipp.erhart@med.uni-heidelberg.de (P.E.); 2Department of Vascular Surgery, Lucerne Cantonal Hospital, 6000 Lucerne, Switzerland; maani.hakimi@luks.ch

**Keywords:** aortic aneurysm, arterial aneurysm, aneurysm distribution, gender, screening

## Abstract

The aim of this study was to investigate sex-dependent aneurysm distributions. A total of 3107 patients with arterial aneurysms were diagnosed from 2006 to 2016. Patients with anything other than true aneurysms, hereditary connective tissue disorders or vasculitides (*n* = 918) were excluded. Affected arterial sites and age at first aneurysm diagnosis were compared between women and men by an unpaired two-tailed t-test and Fisher’s exact test. The study sample consisted of 2189 patients, of whom 1873 were men (85.6%) and 316 women (14.4%) (ratio *m*:*w* = 5.9:1). Men had considerably more aneurysms in the abdominal aorta (83.4% vs. 71.1%; *p* < 0.001), common iliac artery (28.7% vs. 8.9%; *p* < 0.001), internal iliac artery (6.6% vs. 1.3%; *p* < 0.001) and popliteal artery (11.1% vs. 2.5%; *p* < 0.001). In contrast, women had a higher proportion of aneurysms in the ascending aorta (4.4% vs. 10.8%; *p* < 0.001), descending aorta (11.1% vs. 36.4%; *p* < 0.001), splenic artery (0.9% vs. 5.1%; *p* < 0.001) and renal artery (0.8% vs. 6.0%; *p* < 0.001). Age at disease onset and further aneurysm distribution showed no considerable difference. The infrarenal segment might be considered a natural border for aneurysm formation in men and women suspected to have distinct genetic, pathophysiologic and ontogenetic factors. Screening modalities for women at risk might need further adjustment, particularly thoracic cross-sectional imaging complementation.

## 1. Introduction

As a chronic disease, arterial aneurysm formation mostly remains unnoticed and clinically inapparent. Aneurysms are either diagnosed with screening methods or in the context of symptomatic manifestation such as pain, rupture or peripheral embolization. The age-standardized death rate of aortic aneurysms in 2017 was 2.19 of 100,000 persons, which continues to represent a relevant public health issue worldwide [1]. While the abdominal aorta represents the most frequently affected aneurysm site, visceral artery aneurysms account for only 5% of the intraabdominal aneurysms, with reported incidences of 0.01–0.2% [2,3]. Popliteal artery aneurysms represent the most common peripheral aneurysms and, together with femoral artery aneurysms, show a combined prevalence of 7.4 per 100,000 males and 1.0 per 100,000 females in hospitalized patients [4]. Early detection and screening programs are necessary for individual risk stratification. To improve screening efficiency, knowledge of aneurysm frequency, localization and growth rates is essential. As ultrasound is used as the primary aneurysm screening method, certain arterial locations such as the thoracic aorta have limited accessibility and might be underdiagnosed.

Sex differences in the pathogenesis and manifestation of arterial aneurysms and cardiovascular diseases in general are becoming increasingly recognized [5]. In addition to prevalence, men and women show differences in treatment outcomes and natural courses of aneurysmal diseases [6,7]. Although abdominal aortic aneurysms (AAA) are less prevalent in women, aneurysm-related complications, such as faster growth rates and rupture, are higher compared with men [6,8,9]. Similarly, thoracic aortic aneurysms (TAA) are less common in women but have higher rates of growth and rupture compared to men, with a 40% increase in mortality [7,10]. Hormonal, genetic and anatomical differences are assumed as underlying factors [5].

The anticipation of aneurysm formation is essential for preventing aneurysm-related complications. As published for AAA, women had a twofold increased risk for synchronous TAA compared to men [11]. Further arterial aneurysm localization and co-prevalence need to be investigated to evaluate current screening recommendations. In this retrospective, explorative study, we analyzed in detail sex-dependent differences in arterial aneurysm distribution.

## 2. Materials and Methods

Included were all patients diagnosed with an arterial aneurysm between 2006 and 2016 at the Department of Vascular and Endovascular Surgery of the Heidelberg University Hospital. Patients were identified by the ICD-10 classification system (code I71 and I72) from the hospital documentation system. Both patients undergoing aneurysm repair and conservative treatment according to the respective aneurysm guideline were included. Clinical and imaging data of all medical records were assessed. Patients with (1) other arterial pathologies than true aneurysms, such as false aneurysms, dissections or penetrating aortic ulcers and (2) patients with diagnosed connective tissue diseases or acute vasculitides were excluded. Dilated arterial segments were assessed according to the recent respective guideline [2,12,13,14]. Aneurysm localization was compared between men (m) and women (w). In addition, the total number of aneurysms, age at diagnosis of the first aneurysm and cardiovascular risk factors of each patient were assessed.

Number of aneurysms and age at initial diagnosis were compared between sex groups by calculating group-wise mean ± SD and an unpaired two-tailed *t*-test. Aneurysm incidence at each localization and cardiovascular risk factors were compared between sex groups by calculating group-wise incidence, relative risk and Fisher’s exact test. The 95% confidence intervals for relative risk were calculated by the Koopman asymptotic score. All *p*-values are descriptive.

## 3. Results

A total of 3107 patients with arterial aneurysms were diagnosed, and 918 were excluded due to the aforementioned exclusion criteria. In total, 2189 patients with true arterial aneurysms without known confounding systemic disease were used for statistical analysis. Sex distribution was 1873 men (85.6%) and 316 women (14.4%). The sex ratio was 5.93:1 (*m*:*w*).

The abdominal aorta was the arterial segment most frequently affected in both men (*n* = 1562 (83.4%)) and women (*n* = 231 (73.1%)) (Table 1). For the abdominal aorta, as well as other arterial segments, localization patterns were sex-dependent (Figure 1). In men, aneurysms were more frequently located in the abdominal aorta (m: 83.4% vs. w: 73.1%; relative risk (RR): 1.14 (CI 1.07–1.23)), the common iliac artery (m: 28.7% vs. w: 8.9%; RR: 3.24 (CI 2.28–4.66)), the internal iliac artery (m: 6.6% vs. w: 1.3%; RR: 5.23 (CI 2.04–13.60)) and the popliteal artery (m: 11.1% vs. w: 2.5%; RR: 4.39 [CI 2.23–8.72]). In women, a higher proportion of aneurysms could be detected in the ascending aorta (m: 4.4% vs. w: 10.8%; RR: 0.42 (CI 0.28–0.60)), the descending aorta (m: 11.1% vs. w: 36.4%; RR: 0.31 (CI 0.25–0.37)), the splenic artery (m: 0.9% vs. w: 5.1%; RR: 0.18 (CI 0.09–0.35) and the renal artery (m: 0.8% vs. w: 6.0%; RR: 0.13 (CI 0.07–0.26)).

With a mean (±SD) number of synchronous aneurysms of 1.77 ± 1.18 in men and 1.56 ± 0.75 in women, men presented considerably more aneurysms per patient (*p* = 0.003). Men presented a higher proportion regarding diabetes mellitus, dyslipidemia, coronary and peripheral disease. Arterial hypertension and smoking history were similarly distributed in both groups. In combination with an AAA in each case, the most frequent synchronous aneurysm localization in male patients was the common iliac artery (47.4%), the descending aorta (18.7%) and the popliteal artery (10.9%). In contrast, female patients revealed synchronous aneurysms in the descending aorta (64.2%) as most frequent, followed by the common iliac artery (13.2%) and the ascending aorta (10.6%). Women were slightly younger than men, but age of disease onset did not differ considerably (m: 67.52 ± 9.29 y vs. w: 66.39 ± 12.58 y; *p* = 0.117).

## 4. Discussion

This study compared for the first time the distribution of arterial aneurysms between female and male patients. The data show (1) a higher proportion of aneurysms proximally to the infrarenal segment in women and distally in men, (2) a generally higher co-prevalence of synchronous aneurysms in men, and (3) synchronous aneurysms to AAA mainly located in the common iliac artery in men and the descending aorta in women. These results imply that ultrasound screening for arterial aneurysms might detect predisposed arterial regions in men, but additional thoracic cross-sectional imaging should be considered in women.

The higher proportion of arterial aneurysms in the descending aorta in women could explain the aforementioned increased rate of complications despite a lower prevalence than in men. TAA in women—similarly to AAA—is associated with faster growth rates compared to men [7,10].

With a 4:1 ratio, splenic artery aneurysms are reported to be more common in women [15]. Renal artery aneurysms also occur more commonly in women, which is discussed as being associated with higher incidences of fibromuscular dysplasia [16]. Iliac artery aneurysms seem to occur more commonly in men. A recent systematic review on endovascular repair of isolated common iliac artery aneurysms revealed that 90.4% of the patients were male [17]. Likewise, men are considerably more likely to have popliteal artery aneurysms, with ratios (*m*:*w*) described up to 20:1 [18]. Similar to the results of this study, Wallinder et al. described for women with AAA the highest proportion of synchronous aneurysms in the thoracic aorta (31%) [19].

Hormonal and genetic influences in aneurysm formation concerning sex differences are mainly described for AAA. A protective effect of estrogen is presumed by inhibiting matrix metalloprotease (MMP) 9 activity and therefore degradation of the arterial wall [5,20]. AAA growth rates were lower in patients with increased estrogen receptor alpha in the abdominal aortic wall [21]. However, testosterone has the opposite effect: orchidectomy reduced AAA incidence to the level of females in a mouse model [22]. In addition, a protective effect of the X chromosome with regard to aortic aneurysm formation is suspected [23].

Concerning TAA, Sokolis et al. showed increased levels of MMP-2 and MMP-9 in a histopathologic analysis of ascending aortic aneurysms and associated female sex with impaired ascending TAA strength and increased aortic stiffness [24]. Representing a fundamental risk factor of thoracic aortic aneurysm and dissection (TAAD), arterial hypertension is generally more prevalent in female than in male patients of older age [25,26].

Differences in the blood flow patterns depending on the patient’s sex could influence iliac aneurysm formation: In an MRI blood flow study by Taylor et al., only men had diastolic blood flow reversal patterns in the internal iliac artery, probably influenced by the low uterine vascular resistance in women [27]. Accordingly, chronic arterial wall injuries by mechanotransduction are suspected to be higher in men.

Screening of AAA is preferably performed by ultrasound, as for pelvic and lower extremity aneurysms in men [28]. Both the European Society of Cardiology (ESC) and the European Society for Vascular Surgery (ESVS) guidelines currently recommend screening for AAA with an ultrasound scan in all men >65 years of age (Class I, Level A) [13,29]. Attenuated screening recommendations exist for women. Risk factors such as first-degree AAA disease, other arterial aneurysm, tobacco use and lower extremity artery disease should be considered for AAA screening of women.

However, TAA—the descending aorta in particular—are not accessible by ultrasound. Screening of TAA is recommended for high-risk patients, i.e., those with genetic alterations associated with TAAD and first-degree relatives of patients with familial TAAD [30]. While screening of the whole aorta is advocated for AAA prior to surgical intervention, respective recommendations do not exist for smaller AAAs to detect additional thoracic aneurysms [13]. Recently recommended screening modalities seem to be more established in men.

A limitation of this study is the incomplete screening for the whole vasculature, including cranial aneurysms. Although screening modalities in our department are highly standardized following the abovementioned guidelines, some patients might not be screened for any arterial localization.

Furthermore, men and women showed some differences regarding cardiovascular risk factors (diabetes mellitus, dyslipidemia, coronary artery disease and peripheral artery disease). However, decisive risk factors for aneurysm formation, including hypertension and smoking, were comparable between men and women. Additionally, other aneurysm etiologies (hereditary connective tissue diseases and vasculitides) were excluded.

AAA prevalence is 3–4 times higher in men over 65 years compared to women. The male-to-female ratio in our study was 5.9:1, presumably due to the high proportion of iliac aneurysms in our study group, which occur mostly in men.

Since all analyses are based on the population of patients having at least one aneurysm, incidences and effective measures are to be interpreted as conditional. For example, calculated relative risks do not reflect the relative risk between men and women in the total population.

In accordance with the aforementioned guidelines, we recommend AAA screening for peripheral aneurysm patients. In addition, TAA screening should even be considered for female patients with small AAAs. As peripheral and iliac arteries are accessible for ultrasound, CT or MR angiography should be considered in women with AAA to detect synchronous aneurysms of the thoracic aorta, even if AAA has not yet reached a therapy-relevant diameter. In addition, with TAA being more frequent in women, thoracic screening should also be considered early in female patients with aneurysm localizations other than AAA—such as iliac or popliteal. The infrarenal segment could be suspected as a natural border for proportional frequency distributions of aneurysms in men and women.

## 5. Conclusions

Men and women present differences regarding aneurysm localization. After detection of AAAs that are planned for surgical repair or under conservative surveillance, we encourage clinicians to consider iliac and femoropopliteal extended ultrasound screening in men and thoracic cross-sectional imaging complementation in women.

## Figures and Tables

**Figure 1 jcm-11-02450-f001:**
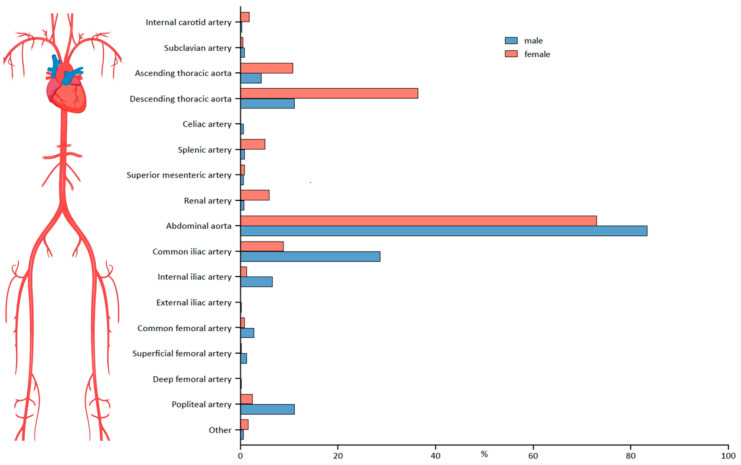
Percentage of aneurysm formation in each arterial segment of men and women. Parts of this figure were created with BioRender.com (accessed on 7 February 2022).

**Table 1 jcm-11-02450-t001:** Characteristics of male and female patients with arterial aneurysm localization.

	Patients	Men	Women	Relative Risk (95% CI)	*p*
Number	2189	1873 (85.6%)	316 (14.4%)	-	-
Mean number of aneurysms ± SD	1.74 ± 1.13	1.77 ± 1.18	1.56 ± 0.75	-	0.003
Mean age at initial diagnosis ± SD	67.37 ± 9.78	67.52 ± 9.29	66.39 ± 12.58	-	0.117
Hypertension	1273 (58.2%)	1091 (58.2%)	182 (57.6%)	1.01 (0.92–1.13)	0.853
Diabetes mellitus	320 (14.6%)	287 (15.3%)	33 (10.4%)	1.47 (1.05–2.07)	0.025
Dyslipidemia	585 (26.7%)	532 (28.4%)	53 (16.8%)	1.69 (1.32–2.20)	<0.001
Smoking	1266 (57.8%)	1098 (58.6%)	168 (53.2%)	1.10 (0.99–1.24)	0.074
Coronary artery disease	633 (28.9%)	566 (30.2%)	67 (21.2%)	1.43 (1.15–1.79)	0.001
Peripheral artery disease	278 (12.7%)	250 (13.3%)	28 (8.9%)	1.51 (1.05–2.19)	0.028
Ascending thoracic aorta	117 (5.3%)	83 (4.4%)	34 (10.8%)	0.42 (0.28–0.60)	<0.001
Internal carotid artery	13 (0.6%)	7 (0.4%)	6 (1.9%)	0.20 (0.07–0.56)	0.006
Subclavian artery	18 (0.8%)	16 (0.9%)	2 (0.6%)	1.35 (0.35–5.26)	1.000
Descending thoracic aorta	323 (14.8%)	208 (11.1%)	115 (36.4%)	0.31 (0.25–0.37)	<0.001
Celiac artery	14 (0.6%)	14 (0.7%)	0 (0.0%)	-	0.243
Splenic artery	33 (1.5%)	17 (0.9%)	16 (5.1%)	0.18 (0.09–0.35)	<0.001
Superior mesenteric artery	16 (0.7%)	13 (0.7%)	3 (0.9%)	0.73 (0.23–2.39)	0.717
Renal artery	34 (1.6%)	15 (0.8%)	19 (6.0%)	0.13 (0.07–0.26)	<0.001
Abdominal aorta	1793 (81.9%)	1562 (83.4%)	231 (73.1%)	1.14 (1.07–1.23)	<0.001
Common iliac artery	565 (25.8%)	537 (28.7%)	28 (8.9%)	3.24 (2.28–4.66)	<0.001
Internal iliac artery	128 (5.8%)	124 (6.6%)	4 (1.3%)	5.23 (2.04–13.60)	<0.001
External iliac artery	5 (0.2%)	5 (0.3%)	0 (0.0%)	-	1.000
Common femoral artery	58 (2.6%)	55 (2.9%)	3 (0.9%)	3.09 (1.04–9.32)	0.038
Superficial femoral artery	26 (1.2%)	25 (1.3%)	1 (0.3%)	4.22 (0.73–24.54)	0.161
Deep femoral artery	4 (0.2%)	4 (0.2%)	0 (0.0%)	-	1.000
Popliteal artery	216 (9.9%)	208 (11.1%)	8 (2.5%)	4.39 (2.23–8.72)	<0.001
Other	19 (0.9%)	14 (0.7%)	5 (1.6%)	0.47 (0.18–1.26)	0.178

Number, mean age at initial diagnosis and cardiovascular risk factors of patients with arterial aneurysms with mean number and localization of arterial aneurysms of all patients, men and women. Mean number of aneurysms and mean age at initial diagnosis by unpaired two-tailed t-test; cardiovascular risk factors and arterial segments by Fisher’s exact test. Relative risk was not calculated if one of the groups was empty.

## Data Availability

Data are available in the manuscript and on personal request to the corresponding author.

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
