# Peer review of "Arterial Aneurysm Localization Is Sex-Dependent"

_jcm, 2022, doi:10.3390/jcm11092450_

Round 1

Reviewer 1 Report

The authors present an easy study to interpret. However, what does this study add? They confirm already established international guidelines.

Title. It is nice to see the use of “sex” and not “gender” in the main title.

Introduction. There is no point in adding reference no. 5. Reference 11 refers to acute aortic dissection. Aortic aneurysm and aortic dissection are two different diseases. Dissection can occur in aneurysms, and chronic dissections can dilate to form aneurysms. However, it is essential to keep the “dissection”-part out of this introduction – along with references 11 and 12. The authors already exclude people with dissections.

It would be more relevant to refer to international guidelines such as ESVS or JVS for arterial aneurysms rather than reference no. 14 from 1991. The authors already refer to JVS Visceral aneurysms.

Remove the metachronous in line 55. The authors do not investigate this in the current study.

Method: It would be preferable with more information regarding the study population. Are the patients treated or followed at the department or both? Are they treated/followed for all aneurysms or only one?

Do the authors have 2189 patients without known confounding factors for aneurysms? They have medical records, so they should mention relevant variables such as hypertension, smoking, and peripheral arterial disease. It is highly relevant to take these factors into account.

I cannot entirely agree with the relative risk and odds ratios used in the analyses. RR is used in cohort studies, while OR is mainly used in case-control studies. This study is neither. Including OR and RR in table 1 adds nothing to the results; it only confuses, especially since the authors do not comment upon the OR.

Line 85: AAA prevalence for women is 73.1% in Table 1 and line 83.

It could be interesting to have a table or an overview of the initial aneurysm and the incidental finding. Furthermore, as mentioned as a limitation in lines 167-169, some overview of how many has complete/incomplete imaging. Is it the thoracic aorta or popliteal segment which is most commonly neglected? Did the authors have a full view of the entire aorta in all cases to comment on the synchronous aneurysms in the descending aorta? And clinical exams or imaging to reveal any popliteal aneurysms?

Figure 1. The authors mention Fisher’s exact test. However, the figure only shows the percentage and no comparisons. The figure is redundant when the authors have the exact numbers in Table 1. Furthermore, Table 1 is easier to read since one does not have any difficulties finding the correct number as in the figure.

Line 120-123. The authors imply additional imaging. However, this is already recommended in the ESVS guidelines: “As the presence of synchronous aneurysms in other vascular beds may influence surgical decision making, screening of the whole aorta and the femoropopliteal segment is advocated.”, page 25, 4.4.1. The authors might consider rephrasing and adding a reference.

Line 126. AAAs are also associated with faster growth rates in women – as mentioned in lines 47-48.

Line 127. The 4:1 ratio; is that number from the reference or this study. Please rephrase if the reference supports the current study.

Line 127-135. There are many locations for aneurysms in a few lines, which makes it somewhat confusing. It is unnecessary to mention all locations in only a few lines – or use more space and elaborate. The same goes for lines 136-153. This study notices a difference in sex and prevalence, but it does not investigate the underlying cause. The hours put into the reference search are priceworthy, but is it recital of many potential underlying causes and new terms in this study (not mentioned in the introduction).

Line 160-161. If the ESVS guidelines recommend screening at ages >65, and the mean age in this study at initial diagnosis is 67 years, why do the authors recommend screening at ages> 60 and not 65?

Line 169-170. This is the first mention of metachronous aneurysms—unnecessary limitation.

Line 172-175. Relevant limitation.

Line 176-185. Is this not a standard procedure at the author’s department? It is recommended – as mentioned above – in the ESVS guidelines. Could the authors extract and emphasise anything from their study in conclusion other than an already recommended message?

Reviewer 2 Report

The Authors proposed an interesting study with the aim to investigate the relation between sex and arterial aneurysm localization. 

The topic is interesting even if the emerged results are in line with well-known data. 

By the way, I have two major concerns:
1- the significant disparity in the numerosity of the two study groups (men and women). This could be something to address in the limitation section.

2-the text completely lacks baseline patients characteristics. Anamnestic features (i.e. cardiovascular risk factors) could have influenced the distribution of aneurysmal lesions. I think that is mandatory to add this data to fully understand the results that the authors have proposed. 

Round 2

Reviewer 1 Report

The authors have improved their manuscript and adressed my concerns. 

Reviewer 2 Report

The previous round of revision has improved the manuscript significantly.